# Parturition and Its Relationship with Stillbirths and Asphyxiated Piglets

**DOI:** 10.3390/ani9110885

**Published:** 2019-10-31

**Authors:** Pieter Langendijk, Kate Plush

**Affiliations:** 1Trouw Nutrition R&D, Stationsstraat 77, 3811 MH Amersfoort, The Netherlands; 2SunPork Group, Murarrie, QLD 4172, Australia; kate.plush@sunporkfarms.com.au

**Keywords:** sow parturition, stillbirth, piglet asphyxia

## Abstract

**Simple Summary:**

Piglets that experience a long labour are more likely to die during birth or immediately following birth, or to grow slowly during life. This is because the longer the piglet experiences contractions during labour, blood supply and oxygen delivery to the brain will be impaired. Even before the first piglet in the litter is born, sows that will have a delayed labour can be identified. This means that the key to saving piglets that are at risk of death because of long labours lies in managing the sow before she gives birth.

**Abstract:**

The transition from an intra- to extrauterine existence is extremely challenging for the pig. This is evidenced by the fact that conservative estimates place intrapartum piglet death at between 5% and 10%. The main cause of this loss is the parturition process itself, with a long farrowing duration resulting in reduced oxygenation to the piglet via uterine contractions stretching, and in some cases, causing rupture of the umbilical cord. Sows that experience a long expulsive stage of parturition are likely compromised before the birth of the first piglet, and so any strategy to reduce stillbirth should be applied before this. Even in piglets born alive, 15% to 20% will have suffered asphyxiation because of a long cumulative farrowing duration. These individuals are significantly disadvantaged with regards to behavioural progression, colostrum intake, growth and survival extending past the lactation phase, and so require more labour and resources than littermates in order to make them viable. Given these immediate and longer-term impacts, identifying ways to correctly manage the sow before parturition leading to a reduction in farrowing duration should be a priority in order to maximise piglet performance.

## 1. Introduction

The transition from intra- to extrauterine life is arguably one of the most challenging phases for the pig. The focus of this review is to describe the physiological processes that the neonate is subjected to around the time of birth and to discuss the suitability of interventions that assist in the transition from inside the uterus to extrauterine life, with a focus on those who experience birth asphyxia. The impacts of birth trauma can be grouped into two specific phases; intrapartum death in which the piglet is stillborn and postnatal morbidity and mortality whereby the piglet is born alive but has experienced asphyxia. These two phases will be detailed in this review, mainly focusing on how the piglet is affected going through the parturition process and neonatal life. The review is limited to physiological conditions of asphyxia and therefore does not cover infectious causes of stillbirth.

## 2. Incidence of Stillbirth

According to recent publications, stillbirth rate in pigs varies between 5% to 10% [1,2,3,4,5], however, in some high prolific herds stillbirth may well be as high as 14%. For example, Vanderhaeghe, et al. [5] reported a range in stillbirth rate of 1.5% to 19.3% in 107 herds. Some stillbirth originates before the onset of parturition, including mummified foetuses and non-fresh stillborns, however, the majority (>75%) of stillborns die intrapartum, due to oxygen insufficiency [6]. The impact of intrapartum events on stillbirth was illustrated by a series of over 800 caesarean sections, where only 1.9% of the piglets were delivered dead. This work suggested that only around 2% of stillbirths are inevitable, these piglets being immature or otherwise non-viable [7]. It is, therefore, the process of parturition itself that poses the risk of stillbirth, not so much the piglet. This is somehow reflected in the similar weight distribution of liveborn and stillborn piglets (Figure 1; unpublished data), suggesting that stillborn piglets are a cross section of the overall population, rather than a category with reduced odds of surviving through parturition. This also to some extent answers the question as to whether reducing stillbirth results in salvaging piglets that are vital enough to perform equal to their littermates. Based on their birthweight distribution, these piglets may be expected to be as viable as their litter mates.

## 3. Parturition and Neonatal Oxygenation

Oxygen deprivation during parturition, as the main cause of stillbirth, may be caused by stretching and even rupture of the umbilical cord during stage II (expulsion of piglets) of parturition [8]. However, the cumulative effect of repeated contractions compressing the placenta and reducing blood flow to the foetus is probably more important [9]. Obstruction of blood flow to the foetus induces a temporary drop in heart rate, which in itself is probably harmless and an adaptive mechanism of the foetus to reduce its oxygen consumption. Repeated episodes of obstructed blood flow, however, can result in anaerobic metabolism, the cumulative effect of which is an increasing lactate level in foetal blood and a decreasing foetal blood pH as buffering capacity is exhausted. This was demonstrated by van Dijk, et al. [10], who clamped the umbilical cord of unborn piglets for 5–8 min during caesarean section, to simulate the cumulative effect of oxygen insufficiency. In observational studies with no interventions, the cumulative effect of oxygen deprivation is reflected in the profound effect of birth order on asphyxia symptoms [10,11], with piglets later in the birth order showing increasingly higher lactate and lower pH in umbilical cord samples at birth. In a more recent study, the total duration of farrowing was shown to exacerbate the effect of birth order [12] on measures of asphyxia and on the risk of stillbirth. This explains why stillbirth rate increases with birth order (Figure 2), especially in sows that experience a prolonged duration of farrowing. From this, it is questionable whether birth intervals contribute to stillbirth, as is sometimes suggested. In fact, a recent study showed that risk of stillbirth only started to increase significantly when birth interval exceeded 90 min, whereas the duration of farrowing increased the risk cumulatively with every 2 h that elapsed [13]. The importance of the cumulative effect of contractions is supported by observations from Zerobin and Kündig [14], showing that inhibition of contractions at the start or just before parturition, using a tocolytic, prolonged parturition without increasing stillbirths.

It is not clear how exactly the mechanics of foetal expulsion work in the pig [15]. Both tubo-cervical and cervico-tubal contractions occur during parturition, the first obviously serving to transport foetuses to the birth canal. The function of the latter is less clear, but have been proposed to shorten the uterine horn and to prevent accumulation of foetuses at the caudal end, thus reducing the premature rupture of the umbilical cord [15]. In this respect, we do not know whether before being expelled, foetuses gradually travel through the uterine horn, or remain positioned at their implantation site and only travel to the birth canal once it is their turn. These two scenarios have different implications for the oxygenation for the foetus. In prolonged parturitions, placentae may also start to detach, since van Rens and van der Lende [16] reported that in contrast to short parturitions where placentae are expelled after the last piglet has been born, in prolonged parturitions the first set of membranes are expelled while some piglets are still to be born.

With the cumulative duration of farrowing and birth order being the predominant factors determining the risk of stillbirth, other piglet-related factors that increase this risk are a ruptured umbilical cord, breech position and meconium staining. It is interesting to note that the percentage of piglets born with a ruptured umbilical cord varies widely from around 20% to 70% [8,12,17], and also that in some studies this percentage increases with birth order [8], whereas in others it is constant [12]. Around 50% or more of stillbirths are associated with a ruptured umbilical cord, which suggests that this is an important risk factor. However, when expressed as the relative risk, a ruptured umbilical cord increases the risk of stillbirth approximately three-fold [12,17]. This is much less compared to birth order and length of parturition. In short parturitions (<3 h for example), stillbirth increased ten-fold between the first and the last piglets in the birth order; in long parturitions, stillbirth increased forty-fold (from 1% to 37%) between the first and the last piglets. The same applies to meconium staining which occurs frequently and increases the relative risk by around three-fold [17]. With meconium staining it is also difficult to assess whether it is a consequence of asphyxia, or whether it contributed to the factors leading up to stillbirth. According to Randall [8], posterior delivery is a physiological condition in pigs and not associated with increased risk of stillbirth.

## 4. Understanding Compromised Parturition

From the above, for improving piglet survival it appears crucial to understand the physiology of the farrowing process and factors that contribute to the length of the farrowing. Towards the end of gestation, secretion of oestrogens by the foetuses trigger the release of prostaglandins, initiating the regression of the corpora lutea, which then cease to secrete progesterone and start to secrete relaxin. These endocrinological changes are typical for parturition stage I, the preparatory phase of parturition, when vulvar swelling, mammary gland fill and dilation of the cervix take place. Before sows go into stage II of parturition, the actual expulsion of foetuses, oxytocin secretion augments and is responsible for the ejection of colostrum a few hours before birth of the first piglet. These changes are fundamental, since recent data have shown that sows that take long to farrow their entire litter also take longer to farrow the first few piglets. This correlation suggests that sows with prolonged parturition are already compromised at the start of parturition and that some of the underlying causes can be found prior to stage II. For example, Rootwelt, et al. [17] showed that delaying the dilation phase by inhibiting or attenuating the activity of relaxin prolonged the duration of subsequent parturition and increased the number of stillborn. Similarly, data from our lab suggest that when stage I is defined as the time between the decline in progesterone and birth of the first piglet, a longer stage I is related to a longer stage II of farrowing [18].

The relevance of stage I of parturition is also evident from the ability of the sow to express nest building behaviour. Various papers have shown that providing substrate that allows the sow to perform behaviour related to nest building, as well as housing conditions that provide more freedom to move around farrowing, have been shown to increase oxytocin around farrowing and to reduce the duration of farrowing and the number of stillborn piglets [3,19]. The importance of oxytocin secretion during farrowing is evident from early studies [20], where both moving a sow during the farrowing process or administration of endogenous opiates severely reduced secretion of oxytocin and disrupted the expulsion of piglets for at least 2 hours. This illustrates the impact a negative stressor can have on the ongoing farrowing process, but the work with pre-farrowing housing conditions equally illustrates the significance of stage I events and housing conditions for dilation and onset of farrowing. These observations are interesting considering that the majority of sows farrow in confined conditions in crates, and these have shown to increase stillbirth rate when compared to pens where sows have more freedom to move and perform nest-building type behaviour [21]. It is also interesting to question how the change from gestation stalls to group housing in gestation affects the transition period. Coming from group housing in gestation, the move to confinement in a farrowing crate may be perceived by the sows even more as a constraint than previously when sows were housed in gestation stalls [22], and this is something that future management should take into account.

Another source of stress around parturition is probably the pain or discomfort associated with constipation. Depending on feed allowance and the fibre content of the diet during the transition phase, as well as water intake, sows may develop constipation. A number of studies have shown that increasing the fibre content, from around 4% crude fibre to around 10% crude fibre, increases water intake in sows before farrowing and also reduces the risk of developing constipation, as measured by the faecal hardness scores or moisture content of the faeces [23]. These studies have also shown that on a high fibre diet, duration of farrowing may be shortened and stillbirth may be reduced. In addition, colostrum intake of low birthweight piglets is increased when their dams are provided a fibrous diet before farrowing [24]. This probably reflects the increased vitality of these piglets as a consequence of an improved farrowing process [23]. These observations are interesting considering that the predominant strategy during transition is to reduce feed allowance and at the same time switch to a lactating sow diet, which generally contains a lower fibre content. The total amount of fibre intake as a result is dramatically reduced, which would presumably aggravate the risk of developing constipation. In the context of feeding strategy prior to farrowing, Feyera, et al. [25] reported that farrowing duration increased with the time since the last meal was ingested, and suggested that glucose levels in the blood of the sow were key to this relationship. Obviously, from the above, a longer farrowing duration increases the risk of stillbirth and therefore, strategies involving a reduction in feed allowance prior to farrowing are questionable. Other ways to improve constipation around parturition are the use of certain yeasts. Tan, et al. [26], for example, used *Sacharomyces boulardii* to improve faecal consistency around farrowing.

A number of studies have attempted to elucidate metabolic changes in the sow that may explain the variation in the duration of farrowing. Le Cozler, et al. [27] provided some insight into the metabolic changes around parturition. They observed a decline in insulin and urea and an increase in non-esterified fatty acids (NEFA) on the day of parturition, probably reflecting fasting or reduced feed intake around farrowing. Glucose levels were unchanged, an observation supported by Oliviero, et al. [23] and Bories, et al. [2]. Cortisol is generally reported to increase around farrowing [19,27]. Le Cozler, et al. [27] also reported an increase in plasma phosphorous and a drop in magnesium on the day of farrowing, probably related to the dephosphorylation of adenosine triphosphate (ATP) in the uterine muscles to provide energy for myometrial activity. They did not observe a change in calcium. Bories, et al. [2] observed differences between sows with a short stage II (173 min average, 0.4 stillborn) and those with a prolonged stage II (297 min average, 1.4 stillborn), and interestingly, these differences were expressed as early as the start of the expulsive phase. Total plasma calcium (Ca) was elevated and magnesium decreased in sows with long parturition, which is confusing but may reflect low uterine muscle activity and therefore low usage of these minerals. Sows with prolonged parturition also had higher total protein levels and lower creatine kinase levels, a difference already established at the start of parturition. Progesterone tended to be higher for sows with a long parturition and tended to be related to a lower colostrum production. Lactate, bicarbonate (HCO_3_) and NEFA were all increased and haematocrit decreased around farrowing, but these levels were not different between sows with short or prolonged parturition. To date, it seems there is there is no clear physiological explanation as to why some sows experience a prolonged stage II, other than reduced oxytocin secretion. It does appear from Bories, et al. [2], and more recently from Langendijk, et al. [12] and Feyera, et al. [25], that sows with prolonged farrowings and increased stillborn are already compromised before the start of farrowing.

Two recent surveys [5,28] pointed to a number of sow- and management-related factors that have not been discussed above. Risk of stillbirth was found to be increased with higher parity sows, presumably due to poor Ca homeostasis or oxytocin secretion in older parities. The authors also found that poor body condition at farrowing (<16 mm back fat) was associated with a higher stillbirth rate and suggested that a poor condition may limit energy available for uterine contractions during farrowing. Interestingly, over-conditioned sows have also been found to be at risk of prolonged farrowing [3]. This emphasises the importance of managing sow body condition in gestation. The history of the sow is also a predictor of stillbirth, as sows with stillborn piglets in previous parturition have a higher risk of giving birth to stillborn piglets. In their survey, Vanderhaeghe, et al. [5] reported that in 7% of parturitions across farms vaginal palpation was applied during farrowing, and that in these cases the stillbirth rate was increased. Whether palpation increases the risk of stillbirth or is a reflection of a sow being at risk already is debatable, and literature on this is equivocal (for review see [5]). Nevertheless, excessive palpation probably aggravates the risk of stillbirth, since it will have the effects of disturbing sows on oxytocin secretion described earlier. In the survey, high prolific genetics had the highest risk of stillbirth. Another interesting observation was the lower stillbirth rate on farms where the ambient temperature in the farrowing house was set below 22 °C [5]. As already discussed, intrapartum piglet death is a major area of economic loss. Perhaps what is less understood is what happens to piglets that experience asphyxia during birth but are born alive. Presumably, the first 24 h are more difficult for these individuals. Incidence, risk factors and impacts, as well as potential interventions to reduce morbidity and mortality will now be discussed.

## 5. Definition and Incidence of Asphyxia in Piglets

There is common agreement on how asphyxiated piglets are identified at birth, with blood biochemical parameter shifts in acid–base balance and oxygenation levels identifiable immediately upon expulsion. To simplify, asphyxiated piglets present with higher partial pressure of carbon dioxide (pCO_2_) and lactate levels, and reduced pH [11] in either circulating or cord blood. Additionally, glucose concentrations may be elevated [11]. Whilst definitive, such biochemical shifts are not always a practical measure of asphyxia in piglets, and so an alternative, more gross indicator is meconium staining [29]. During birth asphyxia, the anal sphincter of the foetus is relaxed and so meconium defecated into the amniotic fluid results in a yellow staining of skin [30]. Piglets with severe meconium staining present with high lactate and glucose, and low pH in cord blood [31,32] suggesting this simple visual score can be used in place of blood biochemical analyses for asphyxia diagnosis.

The frequency of intrapartum death is commonly reported and has been discussed in detail in previous paragraphs. Information on the incidence of asphyxiated piglets that are born alive however is scarce. In Herpin, et al. [11], 67 piglets were born alive, and of these, 14 were classified as being highly asphyxiated, meaning that in this small study, the incidence of asphyxia was 36%. This may be an over-estimation however, as a review by Mota-Rojas, et al. [30] stated that 14% of all liveborn piglets are born with asphyxia-induced low viability. This more conservative estimate is in agreement with Plush, et al. [32], whose 75 severely stained from a total of 386 piglets represented approximately 19% of the population. If we accept that 15%–20% of liveborn piglets are asphyxiated at birth, then this is a considerable number of pigs that will be at a significant disadvantage for reasons that will soon be explored.

## 6. Risk Factors for Birth Trauma

Arguably, the most important risk factor for piglet asphyxia is the process and characteristics of parturition itself. Generally speaking, pCO_2_ is increased with an increasing number of piglets born in the litter [11]. This is however, a rather simplistic view, as in a similar manner already described with respect to stillbirth, pCO_2_ is increased and pH decreased with position in the birth order, but no relationship can be found with birth interval, suggesting impacts on liveborn piglets are cumulative in nature [11]. In support of this, cumulative farrowing duration but not birth interval was shown to be related to degree of meconium staining on liveborn piglets [32]. Those born later in the litter endure greater asphyxia as they experience more uterine contractions, reducing oxygenation via the placenta and increasing the risk of umbilical cord damage [31]. Interestingly, Islas-Fabila, et al. [33] is the only published study in which first born piglets were just as likely to display the physiological indicators of asphyxia as later born piglets. Later born piglets in this study were only classified as 10 to 12 in the birth order, and with litter sizes exceeding this across most parts of the world, this classification of ‘later’ is questionable. Outside of cumulative duration, piglets born in the posterior position [11], and with broken umbilical cords [12] show physiological evidence of asphyxia immediately following parturition.

What is less clear is the relationship between asphyxia and birthweight. Herpin, et al. [11] identified a negative association between blood lactate and weight at birth (i.e., smaller piglets recorded higher lactate levels), and when piglets were classified as asphyxiated or not, the asphyxiated piglets were 25 g lighter. Langendijk, et al. [12] also showed that piglets with the highest lactate concentrations recorded the lowest birthweight (1250 g versus 1530 g). However, others report contrasting findings. Piglets weighing more than 1350 g displayed physiological evidence of asphyxia (low pH and high pCO_2_) and these authors conclude that it is the heavier piglets that are of increased risk of birth trauma [31,34]. In support of this, Bauer, et al. [35] demonstrated that intrauterine growth-restricted piglets (~800 g) were more capable of ensuring brain oxygen demands during asphyxia than normal birthweight piglets (~1500 g). Perhaps, in a similar manner to survival [36], the relationship between piglet birthweight and asphyxia is curvilinear.

Even in the absence of dystocia, sow parity has been reported to influence piglet birth trauma, with first and seventh parity sows giving birth to piglets with higher pCO_2_ levels, lower pH levels and a higher incidence of meconium staining than parity two to six sows [37]. Presumably, the mechanisms behind these parity effects of young and old sows are different. The cervix of the first parity size is small [38], and whilst birthweight is often reduced also [39], foetal pelvic disproportion is the likely explanation for birth trauma in piglets born to sows of this youngest age. Even within parity, cervical size appears to be important [40]. In old sows however, birth trauma is due to reduced oxytocin levels and myometrial tone as discussed with reference to stillbirths in earlier paragraphs. It is then tempting to administer oxytocin in order to reduce farrowing duration with the premise it might reduce piglet asphyxia. However, the incidence of meconium-stained piglets was actually increased when oxytocin was administered after the birth of the first piglet [29]. A subsequent investigation revealed that the use of oxytocin at the birth of the first piglet increased the incidence of severely stained piglets from 21% to 46%. This study was able to demonstrate that exogenous, pharmacological oxytocin increases myometrial activity, decreases foetal cardiac frequency and results in increased umbilical cord breakage frequency [41]. The timing appears important though, as if its use is delayed until the birth of the eighth piglet, the incidence of stained piglets was reduced by 68% [42]. Interestingly, carbetocin, a long acting analogue of oxytocin, resulted in a faster farrowing duration and reduced piglet lactate concentrations when administered after the birth of the first piglet compared with oxytocin-treated and control sows [43]. This alternative may be a more suitable uterotonic than oxytocin to accelerate the birth process whilst reducing piglet asphyxia.

## 7. Asphyxia and Brain Damage

Early investigations determining the pathogenesis of perinatal brain injury utilised piglets as a model for human babies [44]. What was identified, both in the piglet and other experimental species, was that during asphyxia, levels of high energy-containing phosphate compounds in the brain fell during the insult, rebounded quickly after reperfusion and reoxygenation, but then fell permanently by 24–48 h. Thus, severe impacts on individuals surviving birth trauma were likely. This finding was termed ‘secondary energy failure’. Johnston, et al. [45] built on this process by identifying a series of events they referred to as the ‘excito-oxidative cascade’. To summarise their review, this process involves a reduction in ionic gradient across neural membranes and therefore, a reduction in neurotransmitter release. Glutamate re-uptake transporters would ordinarily remove the neurotransmitters from the synaptic cleft, and can work under hypoxemic conditions, but are impaired when ischemia secondary to falling cardiac output restricts the delivery of glucose. Because transporters cannot remove glutamate, it accumulates within synapses, eventually entering the extra-cellular space of the brain. Loss of membrane potential and high glutamate concentrations open the Ca-permeable N-methyl-d-aspartate (NMDA) glutamate channels and voltage-gated Ca channels, causing Ca to move into neurons. Other channels (α-amino-3-hydroxy-5-methyl-4-isoxazolepropionic acid (AMPA) and voltage-dependent calcium channel (VDCC)) are also opened during glutamate binding resulting in further Ca influx. NMDA receptors are upregulated in the neonate because of their role in activity-dependent neuronal plasticity. Also, inhibitory neurotransmitter gamma aminobutyric acid (GABA) results in excitation rather that inhibition in the neonate. It is the Ca flooding into neurons that results in eventual cell death. This is because the enzyme nitric oxide synthetase is activated, leading to high levels of the toxic free radical neurotransmitter nitric oxide. Free radicals then attack enzymes used in oxidative phosphorylation and electron transport. Oxygen free radicals are also generated during re-oxygenation of mitochondria after hypoxia. Ca toxicity results in further attacks to mitochondria from enzymes like caspases, calpains, other proteases and lipases. Damage from mitochondria results in signals causing programed cell death as long as there is energy supply, and during exhaustion of energy supply, necrosis occurs.

## 8. Eventual Outcomes of Asphyxia on Piglets and Management Interventions

By five days of age, it has been reported that almost all of the clinical signs of asphyxia are resolved (with the exception of blood pH and growth [46]), but in this study piglets were housed in thermoneutral conditions and bottle-fed milk replacer, suggesting that when appropriate care is provided, asphyxiated piglets are viable. This then begs the question, how are asphyxiated piglets disadvantaged and what additional care should they receive? After being expelled from the uterus, a piglet must achieve independent enteral feeding and homeothermy to be viable. In order to assess how viable a human neonate is as birth, the APGAR score was developed which incorporates the assessment of breathing effort, heart rate, muscle tone and skin colour. When applied to newborn piglets, those receiving the highest score of 4 had lower blood pCO_2_, higher pH and lower lactate and glucose than those with the lowest liveborn score of 2 [11], indicating that they had suffered less birth asphyxia. In support of this, piglets with severe meconium staining (>40% covering of the skin) recorded a lower viability score at birth [46]. Viability is important as locating the sow’s udder and outcompeting littermates ensures access to colostrum/milk. Time taken for the piglet to reach the udder was reported to increase with increased pCO_2_ and blood lactate as well as with decreased pH [11]; meconium-stained piglets took twice as long to contact the sow’s udder than unstained piglets [46]. Langendijk, et al. [12] showed that asphyxia impacted on the time it took a piglet to successfully feed and most importantly, the volume of colostrum ingested. Piglets with high lactate levels were slower to feed and took in less colostrum than those without evidence of asphyxia. The measurement of such behaviours is important given the link between colostrum intake and piglet survival [47]. Catecholamines are released to stimulate liver glycogenolysis after asphyxia [11]. Because of this, piglets that are asphyxiated [11,32] and that die during the first ten days [48] display elevated blood glucose levels. So, taken collectively, asphyxiated piglets have lower vitality which will compromise feeding and mobilise glycogen stores at a rapid rate, making hypoglycaemia likely. Therefore, it is imperative to restore glucose levels in asphyxiated piglets immediately and employ colostrum management strategies such as those outlined by Alexopoulos, et al. [47].

Asphyxiated piglets also struggle with regards to temperature homeostasis. Heavily meconium-stained piglets recorded rectal temperatures 1.7 °C cooler at birth than those with little to no staining [46]. Rectal temperature at 24 h of age has also shown to increase with pH and with decreased pCO_2_ and lactate [11], and so the negative effect of birth trauma on temperature is maintained in the days following birth. However, Herpin, et al. [49] suggested that it is not homeothermy per se that is impaired in asphyxiated piglets, but rather the link between birth trauma, colostrum intake and carbohydrate metabolism (explored in the previous paragraph) that results in the low body temperature. Regardless, with reduced rectal temperature, these piglets are at increased risk of chilling. Thoresen, et al. [50] identified that secondary energy failure was reduced when piglets were cooled to 35 °C for 12 h after induced asphyxia, with this finding and others being responsible for the clinical application of therapeutic hypothermia for treatment of human neonatal hypoxic-ischemic encephalopathy. Thus, the commonly reported hypothermia in asphyxiated piglets may have neuroprotective effects. How to manage the rewarming event then becomes contentious. Wang, et al. [51] identified increased apoptosis when rewarming was rapid rather than slow (i.e., 2 h versus 8 h), and so a more gradual increase in body temperature of chilled piglets would result in the highest success rate.

Given these impacts on a piglets ability to thermoregulate and ingest colostrum, perhaps it is not surprising that when no additional care is provided, mortality in the first week of life is doubled from 5.5% to almost 11% in asphyxiated piglets [12]. In addition to survival being reduced, piglet growth is compromised when birth trauma is experienced. When piglets are exposed to asphyxia, blood flow is reduced to the small and large intestines during the event, which resulted in intestinal lesions at 12 h of age [52]. Such damage to the gastro-intestinal tract would have ramifications for milk and colostrum absorption. Indeed, piglets with heavy meconium staining gained 168 g by day 5, compared with 520 g for control piglets [46]. By ten days of life, average daily gain (ADG) was shown to be inversely related to blood lactate levels at birth [11]. At weaning, ADG was reduced from 259 g/d to 245 g/d in least versus most asphyxiated piglets, respectively; from weaning to 10 weeks ADG was reduced from 721 g/d to 664 g/d in least versus most asphyxiated piglets, respectively [12]. From these studies, it is evident that asphyxiated piglets perform poorly with regards to growth not only in the farrowing house, but after the weaning event also. What is less clear is the mechanism behind the reduced growth rates. Is it simply that the less viable pigs ingested low colostrum levels (with colostrum intake influencing growth to six weeks of age [53]), or are there longer term impacts of asphyxia on the physiology and development of the pig? Ralph, et al. [54] demonstrated behavioural differences at the end of lactation (day 21) in piglets heavily stained with meconium suggestive of neurological shifts that persist outside the perinatal period. Further work examining whether there are long term gastro-intestinal changes in piglets experiencing asphyxia are warranted.

## 9. Conclusions

Using conservative estimates from this review, 10% of piglets are stillborn and 15% of those born alive are at an increased risk of significant morbidity and mortality from birth trauma. Assuming a total litter size of 15, this equates to 3–4 viable piglets per litter that will be affected by asphyxia, a figure that will only rise with increased prolificacy. The parturition process is almost exclusively responsible, with cumulative farrowing duration rather than birth interval explaining most of the damage to piglets. Interestingly, sows at risk of delivering asphyxiated piglets are already compromised before the expulsive phase. This means that interventions that are applied during the second phase when piglets are birthed will have little impact, and in some cases, may actually increase the odds of a dead or low-viability piglet. Asphyxiated piglets require special care immediately after birth as the condition impairs glucose and temperature homeostasis. Identifying ways to correctly manage the sow before parturition leading to a reduction in farrowing duration should be a priority in order to maximise piglet performance.

## Figures and Tables

**Figure 1 animals-09-00885-f001:**
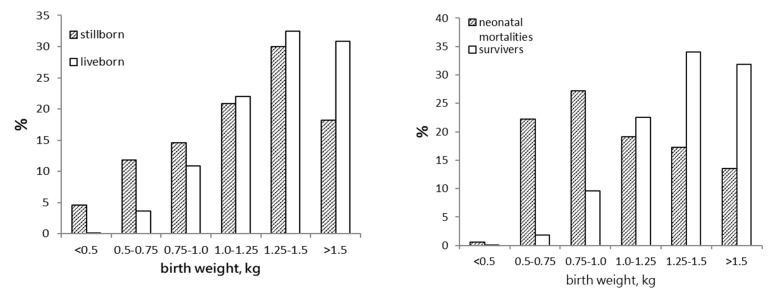
Distribution over different birthweight classes for stillborn piglets and liveborn piglets that survived to weaning (**left panel**) and neonatal mortalities and liveborn piglets that survived to weaning (**right panel**). The percentage on each Y-axis represents the percentage of piglets in each category (stillborn, neonatal mortalities, or liveborn piglets that survive to weaning) that fall into a particular birthweight class. In total, 1856 piglets were recorded (unpublished data). The data show that stillborn piglets and piglets that survive to weaning have a similar birthweight distribution, but that neonatal mortalities are skewed towards low birthweights.

**Figure 2 animals-09-00885-f002:**
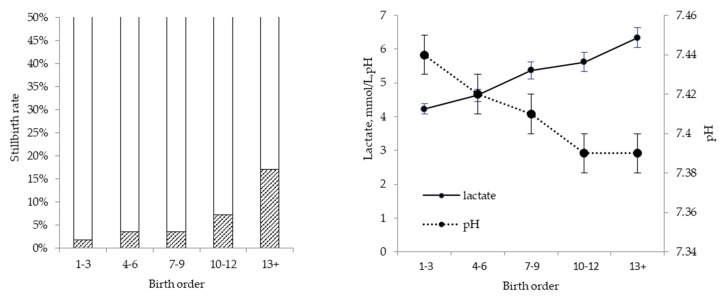
Effect of birth order on stillbirth (**left panel**) and measures of asphyxia (**right panel**) [12].

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
