# Peer review of "Parturition and Its Relationship with Stillbirths and Asphyxiated Piglets"

_animals, 2019, doi:10.3390/ani9110885_

Round 1

Reviewer 1 Report

This is a topic that has been researched and reviewed extensively and so it is hard to provide a fresh discussion of stillbirths in swine. It is interesting that this paper emphasizes the link between stillbirth and piglets born with oxygen deprivation and thus pointing out that the impact of asphyxia is greater than just pigs that are dead at birth. This approach helps make this review somewhat novel. The reference list is extensive and the points made are all valid. However, the difficulty with reviewing such a large topic is that invariably important factors are omitted or given scant coverage. For example, there are several important infectious causes of stillbirth and this topic is not included and its absence is not explained.  Another obvious concern in a literature review is how the emphasis is placed on important factors and not less so on the less important contributing factors. I was surprised at how little attention was paid to age or parity of the sow. I think how the factors were chosen for discussion and the relative importance of each factor needs to be explained.

I think a weakness with some of the studies that have reported the prevalence of stillbirth is that they rely on herd records. There should be some discussion about the studies that have verified whether pigs are stillborn by conducting post-mortem examinations to determine if the pig was born dead or not. On most farms stillbirths include pigs that may have lived for a few hours. It may be useful to clarify the definition and the weakness of farm records in this respect.

Overall this is a well organized literature review with some novel aspects.

Author Response

This is a topic that has been researched and reviewed extensively and so it is hard to provide a fresh discussion of stillbirths in swine. It is interesting that this paper emphasizes the link between stillbirth and piglets born with oxygen deprivation and thus pointing out that the impact of asphyxia is greater than just pigs that are dead at birth. This approach helps make this review somewhat novel. The reference list is extensive and the points made are all valid. However, the difficulty with reviewing such a large topic is that invariably important factors are omitted or given scant coverage. For example, there are several important infectious causes of stillbirth and this topic is not included and its absence is not explained.  Another obvious concern in a literature review is how the emphasis is placed on important factors and not less so on the less important contributing factors. I was surprised at how little attention was paid to age or parity of the sow. I think how the factors were chosen for discussion and the relative importance of each factor needs to be explained.

As the reviewer states, with reviewing such a large topic, invariably some aspects will be omitted whether on purpose or not. In the case of infectious causes, we have purposely decided not included these, as it is another large topic in itself, and we have chosen to limit the review to physiological aspects of parturition. As to factors like parity of the sow, these are mentioned under paragraph 5., but are not discussed in detail because the focus was non the piglet rather than sow or farm related factors. A statement to that effect has been added to the introduction and paragraph 5 begins with mentioning reviews that cover such factors in detail.

I think a weakness with some of the studies that have reported the prevalence of stillbirth is that they rely on herd records. There should be some discussion about the studies that have verified whether pigs are stillborn by conducting post-mortem examinations to determine if the pig was born dead or not. On most farms stillbirths include pigs that may have lived for a few hours. It may be useful to clarify the definition and the weakness of farm records in this respect.

Data on stillbirth prevalence were mainly derived from experimental studies reported in literature, and therefore would be expected to reflect true stillbirth and not include liveborn piglets that dies after birth.

Overall this is a well organized literature review with some novel aspects.

Reviewer 2 Report

Review of the manuscript entitled “Parturition and its relationship with stillbirths and asphyxiated piglets” by P. Langendijk and K. Plush.

General comment

The manuscript is a review that collects and describes key information about two important reasons of losses in piglet production, meaningly stillbirths and asphyxia caused morbidity and mortality. The paper is written proper English language, informative and mostly easy to understand. The greatest value of the article seem to be trial to join the most important information of practical importance and scientific data explaining physiological or even biochemical mechanisms of described phenomena. References are mostly well chosen and used, however, could be filled with  few another new positions. The manuscript is divided into 9 thematic sections excluding Introduction and Conclusion that are another separate sections. The division for such, relatively large number of sections seem to be justified because of large number of factors affecting described phenomena additionally increased by numerous interactions among them. However, the titles of sections sometimes do not fully reflects merit content contained in the text of section. An order of sections also seem to be suboptimal (see specific comments). Generally, the manuscript is valuable set of information in important scientific subject of practical importance and is worthy of publication, however, needs some amendments before acceptance.

Specific comments

Simple summary

Line 10. Add “to” between likely and die

1. Introduction is short section introducing the reader to main subject of paper and justifying its taking. It is well written and needs no corrections.

2. Incidence of stillbirths is a logical continuation of Introduction showing data on frequency of stillborn piglets emergence and simple relationship among this and easy to analyze production parameters. This section contains figure 1 which is a compilation of two charts showing unpublished data describing relationship between birth weight and stillborn piglets, liveborn piglets, neonatal mortality and survival rate. This part of the section raises some doubts. First of all, the description of figure 1 is very general, and the design of data visualisation seem to be difficult to understand and interpret for the first view. In fact the figure itself, without reading the text of section is non informative. The second important doubt is interpretation of the data, suggesting that birth weight is not important for stillbirth (line 47-52). In my opinion the figure in present design do not confirm this interpretation. It is visible, that weight category 1-1.25 is the first where % of liveborn piglets is higher than % of stillbirths, and then, the difference is increasing with growing weight. Careful analysis of figure supported by text reading allow to find that % reflected in Y axis is not related to weight category but to cross section of overall population. Such design is non intuitive according to interpretation of data. The data is unpublished, so I assume this is Author’s property, and could be rebuilt. I would like to see the data in a design where stillborn and liveborn piglets together represent 100% inside every category of birth weight (similarly to fig. 2). In such a design the information should be more proper and easier to clear interpretation in case of birth weight importance.

3. Parturition and neonatal oxygenisation gives information about main physiological mechanisms of oxygen deprivation. This section seem to be OK.

4. Understanding compromised parturition is the longest section of manuscript. It gives key information about physiology of parturition, important for understanding its influence on stillbirth. This section needs some corrections. First of all the title and the text of the section are not fully consistent. There is too much information about housing conditions, management and nutrition of gilts and sows. Those information should be moved to the next section. All thew section should be focused on physiological mechanisms. In physiological description of parturition the effect of cortisol seem to be neglected (the only information about cortisol is one sentence in line 167). This sentence should be moved to the beginning of section and discussed in connection to estradiol, prostaglandin and progesterone secretion. In this subject the paper of Edgerton et al. 1996 should be useful (Biol. Reprod. 55(3), 657-662, 1996).   

Line 113. Add “start to” before secrete relaxin.

Line 144-161. This paragraph should be moved into section 5. This also should be filled with additional information about another methods to prevent constipation like the use of yeast, or popular many years ago KCl. The Authors should use the paper of Tan et al. 2015 (Anim. Feed Sci. Technol. 210, 254-262, 2015).

5. Sow and farm risk factors for intra-partum death. This section contains important information but in very general form. It should be expanded using information about management, housing conditions and nutrition moved from section 4. In this section it would be worthy to mention the effect of differences in reproductive tract size on stillborn and mummified piglets born. The paper of Tuz et al. 2019 should be useful (Animals 9, 158, 2019; doi:10.3390/ani9040158).

Next 3 sections should be changed in order. In my opinion it would be much more logical and better connected if  section 7 would be 6, section 8 would be 7 and section 6 would be 8.

In present section 6 some abbreviations needs to be defined (NMDA, AMPA, VDCC, pCO2). Even if this abbreviations are in wide use, they should be defined during the first use, especially that some other abbreviations in other sections are defined (ATP in line 169 or ADG in line 329).

Line 263. Change ten to 10.

Line 279-285. The relation between size of cervix and number of stillborn piglets can be also confirmed by citation of Tuz et al. 2019 (Animals 9, 158, 2019; doi:10.3390/ani9040158). In this paper the sows were divided into 3 groups according to the size of vagina and cervix, and results showed the influence of this parameter on the number of stillbirths.

9. Eventual outcomes of asphyxia on piglets is the nice try to show possible method and tools to analyse of potential asphyxia in liveborn piglets. In my opinion information from this section could be combined with the next section 10 (Management of affected piglets) as a one larger section.

11. Conclusion seem to be well thought and contain short, but the most important information of practical importance.

To summarize, in my opinion the manuscript is well prepared and worthy of publication after a moderate revision. I hope the Authors will find my comments useful.

Author Response

Specific comments

Simple summary

Line 10.  Add “to” between likely and die

was added

Introduction is short section introducing the reader to main subject of paper and justifying its taking. It is well written and needs no corrections. Incidence of stillbirths is a logical continuation of Introduction showing data on frequency of stillborn piglets emergence and simple relationship among this and easy to analyze production parameters. This section contains figure 1 which is a compilation of two charts showing unpublished data describing relationship between birth weight and stillborn piglets, liveborn piglets, neonatal mortality and survival rate. This part of the section raises some doubts. First of all, the description of figure 1 is very general, and the design of data visualisation seem to be difficult to understand and interpret for the first view. In fact the figure itself, without reading the text of section is non informative. The second important doubt is interpretation of the data, suggesting that birth weight is not important for stillbirth (line 47-52). In my opinion the figure in present design do not confirm this interpretation. It is visible, that weight category 1-1.25 is the first where % of liveborn piglets is higher than % of stillbirths, and then, the difference is increasing with growing weight. Careful analysis of figure supported by text reading allow to find that % reflected in Y axis is not related to weight category but to cross section of overall population. Such design is non intuitive according to interpretation of data. The data is unpublished, so I assume this is Author’s property, and could be rebuilt. I would like to see the data in a design where stillborn and liveborn piglets together represent 100% inside every category of birth weight (similarly to fig. 2). In such a design the information should be more proper and easier to clear interpretation in case of birth weight importance.

We understand the comments from the reviewer, and in hindsight, the description under this figure is poor. We have corrected the description to allow better understanding of the figure. Making the categories of piglets add up to 100% will not convey the message that we want to bring (see below), because it will not show the distribution across weights intuitively.

Parturition and neonatal oxygenisation gives information about main physiological mechanisms of oxygen deprivation. This section seem to be OK. Understanding compromised parturition is the longest section of manuscript. It gives key information about physiology of parturition, important for understanding its influence on stillbirth. This section needs some corrections. First of all the title and the text of the section are not fully consistent. There is too much information about housing conditions, management and nutrition of gilts and sows. Those information should be moved to the next section. All thew section should be focused on physiological mechanisms. In physiological description of parturition the effect of cortisol seem to be neglected (the only information about cortisol is one sentence in line 167). This sentence should be moved to the beginning of section and discussed in connection to estradiol, prostaglandin and progesterone secretion. In this subject the paper of Edgerton et al. 1996 should be useful (Biol. Reprod. 55(3), 657-662, 1996).

We wrote this paragraph in this way to combine physiological mechanisms and practical implications in order to explain impacts of management on parturition. Cortisol is mentioned where it is mentioned because that in that section we sought to identify studies that look at differentiating compromised and non-compromised farrowings. The paper by Edgerton does not refer to parturition.

Line 113. Add “start to” before secrete relaxin.

Was added

Line 144-161. This paragraph should be moved into section 5. This also should be filled with additional information about another methods to prevent constipation like the use of yeast, or popular many years ago KCl. The Authors should use the paper of Tan et al. 2015 (Anim. Feed Sci. Technol. 210, 254-262, 2015).

We believe this section should be located here, because it describes physiological aspects in relation to impacts of practical measures. We agree that in that context pararaph 4. and 5. may as well be merged, and hence we have added the content of paragraph 5 to that of 4.

Sow and farm risk factors for intra-partum death. This section contains important information but in very general form. It should be expanded using information about management, housing conditions and nutrition moved from section 4. In this section it would be worthy to mention the effect of differences in reproductive tract size on stillborn and mummified piglets born. The paper of Tuz et al. 2019 should be useful (Animals 9, 158, 2019; doi:10.3390/ani9040158).

Again, the purpose of the review was to focus on the piglet perspective, and therefore this section was limited to its current size. The work by Tan is now included.

Next 3 sections should be changed in order. In my opinion it would be much more logical and better connected if  section 7 would be 6, section 8 would be 7 and section 6 would be 8.

The authors agree and have made the suggested restructure to these three sections.

In present section 6 some abbreviations needs to be defined (NMDA, AMPA, VDCC, pCO2). Even if this abbreviations are in wide use, they should be defined during the first use, especially that some other abbreviations in other sections are defined (ATP in line 169 or ADG in line 329).

All acronyms have now been defined.

Line 263. Change ten to 10.

The authors could not find a reference to ten at this line number. Ten is written in line 280 however, and this has been changed to 10.

Line 279-285. The relation between size of cervix and number of stillborn piglets can be also confirmed by citation of Tuz et al. 2019 (Animals 9, 158, 2019; doi:10.3390/ani9040158). In this paper the sows were divided into 3 groups according to the size of vagina and cervix, and results showed the influence of this parameter on the number of stillbirths.

Reference added.

Eventual outcomes of asphyxia on piglets is the nice try to show possible method and tools to analyse of potential asphyxia in liveborn piglets. In my opinion information from this section could be combined with the next section 10 (Management of affected piglets) as a one larger section.

We thank the reviewer for their suggestion, and the two sections have now been combined.

Conclusion seem to be well thought and contain short, but the most important information of practical importance.

Reviewer 3 Report

Rarely do I not have a significant amount of input for authors, however; this paper is so well written that I have only a few suggestions/comments. This paper is very well written and is comprehensive in its review of stillbirths/asphyxiation of piglets. It was a very easy read and extremely informative. I'm excited for it to be published so I can share it with other scientists adn farm staff.

The only comments/suggestions I have are as follows:

Sometimes it is not clear if the authors are stating something without reference, or if thier statement refers to the reference that came several sentences/topics prior. For instance, line 201 makes a statement about room temperature and stillbirths. Is the reader to assume that refers to reference #5, which is two sentences prior? I suggest either adding the reference again for certainty, or wording the sentence some how to refer back to that sentence - "That review noted that...."

Line 293, eighth, not eight.

Lines 317-318. As written it sounds like colostrum intake is different than feeding.

Several times the authors mention neurological effects (line 338 for example, and line 354). I think this review would be even more interesting if the authors could expand on this aspect. Specifically how/where is the brain damaged and what are the observable/measurable effects and their implications to production/welfare.

Line 344. 'begs the question.....' but the authors don't provide a succinct answer. I think it would be more useful if the authors listed a, b, c, and d. to answer the question.

Author Response

Sometimes it is not clear if the authors are stating something without reference, or if their statement refers to the reference that came several sentences/topics prior. For instance, line 201 makes a statement about room temperature and stillbirths. Is the reader to assume that refers to reference #5, which is two sentences prior? I suggest either adding the reference again for certainty, or wording the sentence some how to refer back to that sentence - "That review noted that...."

Reference has been repeated.

Line 293, eighth, not eight.

Changed.

Lines 317-318. As written it sounds like colostrum intake is different than feeding.

Changed to ‘’the volume of colostrum ingested’’.

Several times the authors mention neurological effects (line 338 for example, and line 354). I think this review would be even more interesting if the authors could expand on this aspect. Specifically how/where is the brain damaged and what are the observable/measurable effects and their implications to production/welfare.

The etiology of asphyxia on the brain is described in section 6 (now 8 due to other reviewer comments). The authors agree that this is a very interesting area of research, but outside studies looking at neonatal feeding behavior, and our conference abstract which anxiety and boldness were examined (Ralph et al 2015) there is little work done in this area. It would be nice to see some neuro-endocrinological studies, both in the short and long term.

Line 344. 'begs the question.....' but the authors don't provide a succinct answer. I think it would be more useful if the authors listed a, b, c, and d. to answer the question.

This section has now been reworked to combine each impact and management strategy. We feel this has improved clarity.